# Prioritising Climate Change Mitigation Behaviours and Exploring Public Health Co-Benefits: A Delphi Study

**DOI:** 10.3390/ijerph20065094

**Published:** 2023-03-14

**Authors:** Priyanjali Ratwatte, Helena Wehling, Revati Phalkey, Dale Weston

**Affiliations:** 1Behavioural Science and Insights Unit (BSIU), UK Health Security Agency (UKHSA), Porton Down, Wiltshire, Salisbury SP4 0JG, UK; 2Climate Change and Health Unit, UK Health Security Agency (UKHSA), Chilton, Oxon OX11 0RQ, UK; 3Behavioural Science and Evaluation, Health Protection Research Unit (BSE HPRU), Bristol BS8 2BN, UK; 4Environmental Change and Health, Health Protection Research Unit (ECH HPRU), London WC1H 9SH, UK

**Keywords:** climate change mitigation, climate change behaviour, climate change and health, carbon emissions, climate change interventions, anthropogenic climate change, consumption climate change, domestic heating climate change, transportation climate change

## Abstract

Climate change requires urgent action; however, it can be challenging to identify individual-level behaviours that should be prioritised for maximum impact. The study aimed to prioritise climate change mitigation behaviours according to their impacts on climate change and public health, and to identify associated barriers and facilitators—exploring the impact of observed behaviour shifts associated with COVID-19 in the UK. A three-round Delphi study and expert workshop were conducted: An expert panel rated mitigation behaviours impacted by COVID-19 in relation to their importance regarding health impacts and climate change mitigation using a five-point Likert scale. Consensus on the importance of target behaviours was determined by interquartile ranges. In total, seven target behaviours were prioritised: installing double/triple glazing; installing cavity wall insulation; installing solid wall insulation; moving away from meat/emission heavy diets; reducing the number of cars per household; walking shorter journeys; and reducing day/weekend leisure car journeys. Barriers related to the costs associated with performing behaviours and a lack of complementary policy-regulated subsidies. The target behaviours are consistent with recommendations from previous research. To ensure public uptake, interventions should address behavioural facilitators and barriers, dovetail climate change mitigation with health co-benefits and account for the long-term impacts of COVID-19 on these behaviours.

## 1. Introduction

Climate change is a global issue that needs to be addressed urgently via societal-level change, including changes to human behaviour, to limit global warming to 1.5 °C above pre-industrial levels [1]. It is estimated that approximately 150,000 global annual deaths are attributed to climate change, caused directly or indirectly by: extreme weather events; increased transmission of infectious diseases; changes in food production systems; and negative impacts on fossil fuel consumption on air quality [2]. Climate change mitigation behaviours can be defined as behaviours that reduce or prevent greenhouse gas emissions that contribute to ongoing climate change [3]. These climate change mitigation behaviours are commonly split into the following categories: transport behaviours (including surface transport and aviation), domestic heating behaviours and consumption behaviours (electricity consumption and consumption in relation to sustainable diets) [4].

Furthermore, the impacts of climate change-related behaviours could also have consequences that extend beyond the immediate influence on global warming, as many behaviours, for example, active travel or reducing meat consumption, are associated with additional benefits such as improvements in personal health [5]. For example, a heavy reliance on carbon-based fuel transportation indirectly supports sedentary lifestyles, which subsequently leads to an increase in global rates of obesity and related chronic diseases. These adverse outcomes could, however, be reduced by increasing the uptake of climate change mitigation behaviours that support low-carbon active travel where appropriate [6] for example, travelling via bicycle or public transport. Despite the importance of such behaviours for influencing public health, the majority of interventions aimed at promoting the uptake of climate change mitigation behaviours are focused on knowledge provision and present little success in promoting long-term behaviour change [7,8,9,10]. On the other hand, social comparison messaging-based interventions and choice architecture-type interventions are less frequently employed but may be more promising in terms of enabling long-term behaviour change [7,9,10]. Further work is therefore needed to explore and develop effective interventions for increasing climate change mitigation behaviours.

Moreover, despite an upward trend in research examining health aspects of climate change, this area is still significantly behind in comparison with climate-related publications from other sectors [11]. Considering the well-demonstrated link between certain climate change behaviours and health benefits [12], more research is needed to explore those behaviours through the lens of health. Combining climate and human health benefits could also help to generate wider reach by providing people with an engaging and personally relevant frame and could potentially further motivate more individuals to engage in climate change [13].

The COVID-19 pandemic has had a considerable impact, both on population- and individual-level behaviours, mainly as a result of pandemic response measures, with some temporary changes contributing to climate change mitigation and some to exacerbation [14,15,16]; for example, significant behavioural changes have been observed in air and land-based transportation for work and tourism purposes [14,15,16]. This suggests that, despite the catastrophic impacts, both personal and economic, wrought by COVID-19, there is a possibility to build back greener by focusing on developing and implementing behaviour change interventions to promote sustainability that meet greenhouse gas emissions reduction targets as part of COVID-19 recovery [17]. In this way, it may be possible to use the subsequent recovery from the pandemic to help avert future climate change-related crises through sustained behaviour change. To achieve widespread and sustained change, it is important that temporary changes observed in climate mitigation behaviours (for example, behavioural shifts resulting from national lockdowns and other pandemic-related behavioural guidance) are translated into long-term changes that could be sustained beyond the pandemic [14].

While work by McKinnon and colleagues has sought to specify and describe climate change mitigation behaviours in detail [18], this is only the beginning of the endeavour. Further work is needed to both: (a) rank and prioritise these behaviours for further research/intervention development and (b) identify existing work relating to barriers/facilitators/interventions for addressing these behaviours. Such a full consideration and prioritisation of behaviours can help us identify the areas that should be the focus of further work (e.g., those that are priorities but may be relatively understudied, or those where there is plentiful research but there may be less intervention development/testing/evaluation).

In order to facilitate just such a prioritisation, we conducted a multi-stage Delphi study and expert workshop with stakeholders from the UK (see methodology for more details). This process aimed to bring together experts from across behavioural science, public health, and climate change to: reach a consensus concerning priority behaviours and areas for further work; examine the evidence base in relation to relevant target behaviours; and identify existing interventions.

## 2. Materials and Methods

The Delphi method is a quantitative group facilitation technique aimed at generating consensus on issues or subjects by systematically aggregating opinions from a group of experts. The Delphi method is often employed when available knowledge on a subject area is insufficient and other methods that provide higher levels of evidence cannot be used. Consensus is reached by expert ranking processes over several iterative rounds. It requires a minimum of two rounds, the number of rounds is dependent on the timeframe and whether a clear literature base exists about the topic of interest [19,20,21,22,23].

It was decided by our institutional review board that ethical approval was not required as the present research served to explore participants’ perceptions as part of their expert role in the research area on a purely professional level.

The present Delphi study consisted of three iterative rounds. Expert stakeholders were identified through several routes, including contacting existing contacts of the research team and identifying key authors of relevant papers from behavioural science, public health, and climate change organisations/groups. Identified experts were subsequently approached via email with an invitation to participate in the research that summarised the purposes and methodology of the project. To ensure that panel members would be able to provide relevant expertise for the purpose of this study, all eligible participants were required to be:-Technical specialists and or policy makers (e.g., academics, researchers, planners, environmentalists, climatologists, public health experts) who have actively undertaken research (presenting and/or publishing) investigating the behavioural and health impacts of climate change-18 years of age+-Fluent English speakers-UK-based (to identify impacts on climate change mitigation behaviours in the UK).

### 2.1. Round 1: Identification of Target Behaviours

The first round of the Delphi study aimed to identify climate change mitigation behaviours that were impacted by COVID-19 and the associated response measures that were introduced across the UK. This first round comprised an online survey (via the survey platform Select Survey) and commenced on the 27 January 2021. The panel was granted a 10-day period to complete the survey.

The expert panellists were presented a list of pre-determined climate change mitigation behaviours that included 37 behaviours (see Appendix A) that were prioritised according to emissions reduction potential based on the Net Zero Societal Change Analysis Programme by the Department for Business, Energy and Industrial Strategy (BEIS) and Energy Systems Catapult [18]. Following the BEIS approach, the behaviours were classified into the following categories: heat behaviours, transport behaviours, consumption behaviours, electricity behaviours and non-sectoral/societal change behaviours. Participants were initially asked to rate each behaviour on a 3-point scale, indicating whether they thought there had been a ‘positive impact’, a ‘negative impact’ or ‘no impact’ of COVID-19 on each behaviour. The next step involved experts identifying any additional climate change behaviours, impacted by COVID-19, that were not included on the list. Behaviours that were identified as impacted (either positively or negatively) were progressed for rating in round 2, and behaviours that were not identified as being impacted were not progressed for rating.

### 2.2. Round 2: Prioritisation of Target Behaviours

Round 2 of the Delphi focussed on prioritising the climate change mitigation behaviours that were identified as being impacted by COVID-19 in round 1. Participation involved completing an online survey that was emailed to participants, who were then given a period of two weeks to submit their responses. In the survey, experts were asked to rate the importance of each climate change behaviour that was identified from the first round on the basis of two criteria, including (1) climate change impacts and (2) health impacts. For each of these two categories, participants rated each behaviour on a 5-point Likert scale of importance (i.e., 1 = not at all important; 2 = slightly important; 3 = moderately important; 4 = important; 5 = very important). Behaviours that did not reach consensus (see Section 2.3.2) on both criteria (regardless of whether or not they reached consensus on one criteria) were progressed to Round 3 for further rating to try and establish consensus. Behaviours that were identified as important for both criteria were deemed to have reached consensus and did not need to be considered further in Round 3.

In addition, the expert panel was presented with the ‘additional climate change mitigation behaviours’ identified from Round 1 of the study (see Section 2.1) and were asked to rate whether they thought there had been a ‘positive impact’, a ‘negative impact’ or ‘no impact’ of COVID-19 in the UK on each behaviour. The additional behaviours that were identified as impacted (either positively or negatively) were progressed for rating in the next round (Round 3) of the Delphi study, and behaviours that were not identified as being impacted were not progressed for rating.

### 2.3. Round 3: Seeking Consensus on Behaviours That Did Not Reach Consensus Round 2

An expert workshop was facilitated by three members of the research team on the 8th of March 2021 from 9.30 a.m. to 1.30 p.m. with the Delphi expert panellists. The workshop was split into two parts: Round 3 and Discussion. Round 3 is presented within this section, and the Discussion segment is presented in the following section.

In the first part of the workshop, the aim was to discuss behaviours that had not yet reached consensus in order to see whether consensus could be reached ahead of the second part of the workshop. In this session, the results from the previous round were summarised: the coordinators provided an overview of the climate change mitigation behaviours from the second round, which had reached consensus and then presented participants with behaviours that had not yet reached consensus. Participants were then asked to discuss these behaviours in relation to whichever criteria they had not yet reached consensus on (climate change, health, or both). Each behaviour was presented according to its corresponding category along with its calculated importance score. Following this discussion, the expert panel was granted a 15 min period to complete a survey in which they were asked to rate the importance of the climate change mitigation behaviours (that did not reach consensus in Round 2) using the same methodology used in Round 2.

Following the completion of the survey, a rapid analysis of the results to determine consensus was conducted within the 30 min break period to identify target climate change mitigation behaviours. All behaviours that reached consensus (on both criteria) were retained for the Discussion segment of the workshop; all behaviours that did not reach consensus were removed from the process.

#### 2.3.1. Target Behaviour Discussion

In the second part of the workshop, the expert panel was presented with the climate change mitigation behaviours that reached consensus. The discussion focused on understanding the importance of climate change impacts and health impacts, identifying barriers, facilitators and interventions to target behaviours and existing evidence addressing the behaviours. Questions and discussion points can be found in Appendix A.

The panellists were encouraged to focus on specific behaviours if they so desired. The discussion was facilitated using Jamboard, a digital whiteboard platform that served as an interactive discussion tool. Panellists were able to add comments to the board anonymously and were then prompted to discuss them verbally. In addition to the discussion of the importance of climate change behaviours and the barriers and facilitators, the panel was asked to identify future research in the context of the discussed behaviours, and was presented with the following scenario:

‘*If you had one year to conduct research/design and implement an intervention for any of the following behaviours, what would you do?*’

The purpose of this part of the discussion was to generate potential research directions for the forthcoming years of the research team’s projects.

#### 2.3.2. Analysis

Behaviours were considered to have been impacted by COVID-19 in the UK for Round 1 (and additional behaviours introduced in Round 2) if: over 50% of the expert panel rated the behaviour as being impacted by COVID-19 (over 50% rating of ‘positive impact’ or ‘negative impact’ or a combination of both). In Rounds 2 and 3, consensus on the importance of behaviours was calculated by determining the interquartile range (IQR) for the behaviours, respectively, for climate change impacts and health impacts criteria. IQR calculation is an established robust method of consensus determination in Delphi studies because it sets a pre-determined level of consensus prior to analysis [19,20,21,22]. The IQR measures the dispersion of the median and indicates where the middle 50% of observations lie [19]. The IQR is dependent on the number of units on a scale; an IQR of 1 or less is recommended as a consensus indicator for 4- or 5-unit scales. The IQR score is calculated by finding the median value of the lower quartile (Q1) and upper quartile of the data (Q3) and finding the difference between these values (Q3–Q1). An IQR value of less than 1 indicates that over 50% of all opinions lie within a single point on the scale. For the purposes of the current study, consensus measurement was interpreted as follows:-An IQR value of 1 or less (IQR ≤ 1) indicated that consensus had been reached for the behaviour.-An IQR value of higher than 1 (IQR > 1) indicated that consensus had not been achieved for the behaviour.

For a behaviour to be considered to have comprehensively reached consensus, it needed to have achieved consensus for both climate change impacts and health impacts criteria (IQR ≤ 1 for both criteria on a single behaviour). The degree of importance for each behaviour was determined by calculating the median as recommended by Heiko [19].

## 3. Results

### 3.1. Round 1

#### 3.1.1. Impacted Behaviours

A total of 21 expert panellists completed Round 1. Out of these experts, 12 had expertise in the application of behavioural science to climate change and 9 were public health and climate change experts. The behavioural science experts consisted of professors, senior lecturers and research fellows from academic settings (n = 5) and senior policy advisors and researchers from government institutions (n = 7). The areas of expertise covered by the behavioural science expert group were inclusive of: transportation-engagement behavioural policy, plant-based nutritional behavioural policy, environmental psychology, energy efficiency behaviours and low carbon lifestyles. The public health experts consisted of professors, senior lecturers and research fellows from academic settings (n = 5) and senior policy advisors and researchers from government institutions (n = 4). The areas of expertise covered by the public health experts included: indoor environments and high temperatures, built environments and health behavioural policy, climate change and mental health and public health adaptation to climate change.

Although experts indicated whether behaviours were ‘positively impacted’ or ‘negatively impacted’, for the analysis, we summarised negative and positive ratings more broadly as ‘impacted by COVID-19. A total of 20 behaviours were identified as being impacted by COVID-19 in the UK and were progressed to Round 2 for rating. Climate change mitigation behaviours from the ‘heat’ (n = 8), ‘transport’ (n = 5), ‘consumption’ (n = 2) and ‘non-sectoral/societal’ (n = 5) behavioural categories were identified as being impacted. None of the behaviours from the ‘electricity’ behavioural category were identified as being impacted. In total, 17 behaviours that were identified as not being impacted by COVID-19 in the UK were not progressed to Round 2 of the study for rating. The percentage of panellists who identified the respective behaviours as being impacted is included in Table 1.

#### 3.1.2. Additional Behaviours

A total of 20 additional climate change mitigation behaviours were identified by the expert panel as being impacted by COVID-19 in the UK (not included in the pre-determined lists of behaviours in Round 1). These were grouped into the following pre-defined categories created by the research team, including ‘heat’ (n = 5), ‘transport’ (n = 8) and ‘consumption’ (n = 7). No additional climate change mitigation behaviours were specified by the panel for the ‘electricity’ and ‘non-sectoral/societal change’ behavioural categories.

### 3.2. Round 2

#### 3.2.1. Climate Change Mitigation Behaviours Consensus

A total of 20 expert panellists participated in Round 2. A total of three behaviours (installing double/triple glazing; installing cavity wall insulation; installing solid wall insulation) reached consensus from the ‘heat’ behavioural category, a single behaviour from the ‘consumption’ behavioural category (eating healthy/eating more fruit and vegetables; moving away from emission-heavy diets) and two behaviours from the ‘non-sectoral/societal change’ category (looking for jobs in a new green economy; looking for jobs closer to home). None of the behaviours from the ‘transport’ category reached consensus. Behaviours that did not reach consensus across categories can be found in Table 2, including IQR scores, individual importance scores for the respective criteria out of 5 (climate change impact and health impact) and cumulative impact scores (score out of 10 compounded for both criteria) for all behaviours evaluated in Round 2.

#### 3.2.2. Additional Behaviours Impacted by COVID-19

In summary, 17 climate change mitigation behaviours were identified as being impacted by COVID-19 in the UK and were progressed to Round 3 for rating. More specifically, three behaviours from the ‘heat’ category were rated as impacted, and all behaviours in the ‘transport’ (n = 8) and ‘consumption’ (n = 6) categories, respectively, were identified as being impacted. Two behaviours were identified as not being impacted by COVID-19 in the UK and were not progressed to Round 3 of the study for rating. The full list of behaviours and impact percentage scores can be found in Table 3.

### 3.3. Round 3

A total of nine experts participated in Round 3. Three behaviours from the transport behavioural categories reached consensus at the end of Round 3 of the Delphi study, and these included ‘Reducing number of cars per household’, ‘Walking shorter journeys’ and ‘Reducing day/weekend leisure car journeys’. IQR scores, individual importance scores for the respective criteria out of 5 (climate change impact and health impact) and cumulative impact scores (score out of 10 compounded for both criteria) for all behaviours that reached consensus in Round 3 (including additional behaviours rated for COVID-19 impact in Round 2) can be found in Table 4.

The number of behaviours that were included and excluded through each round of the Delphi study can be found in Figure 1.

### 3.4. Expert Stakeholder Workshop

During the workshop discussion, panel experts elected to focus on behaviours from the categories, ‘heat’, ‘consumption’ and ‘travel’. A detailed summary of the discussion and additional literature identified by the panel for each prioritised behaviour can be found below. The research ideas and additional literature resources identified by the panel are listed in Table 5.

#### 3.4.1. Heat Behaviours (Installing Double/Triple Glazing; Installing Cavity Wall Insulation; Installing Solid Wall Insulation)

Whilst panellists felt that heat behaviours were important to promote in the general population, common barriers to adopting heating-insulation home upgrades were related to the high costs associated with home upgrades. Accordingly, facilitators emerging from the discussion were primarily policy related in terms of providing financial incentivisation to facilitate home upgrades and to create compatible, cost-effective building standards. Furthermore, focussing on behaviours related to housing and heat insulation upgrading was considered an effective measure for promoting a behavioural shift in the public. This view stemmed from the panellists’ perceived failure of the recent Green Deal energy efficiency housing scheme, which may have created a sense of urgency amongst policymakers to identify successful methods for encouraging homeowners to adopt more climate-friendly installation systems. Additionally, there was agreement that COVID-19 has indirectly impacted these behaviours. One example, an observable change is that a larger proportion of the UK’s population has been working from home, thus leading to an increase in energy usage for heating. Indeed, participants reflected that emerging data suggest that an increase in home working has not offset the supposed reduction in use from offices or workplaces. Furthermore, as a result of the observed shift towards working from home, more people may be likely to buy larger homes, which may subsequently result in an increased demand for heating amongst private homeowners.

In terms of existing research, two studies were highlighted in relation to the Green Deal, which established policies for enabling energy efficiency behaviours, such as home renovation, and emphasised the importance of certainty of financial benefits for promoting energy-efficient behaviours [24,25]. Moreover, a systematic review describing the impact of teleworking on climate was referenced, which identified a reduction in energy use but concluded that teleworking may lead to unpredictable increases in non-work travel and home energy use that may outweigh the gains from reduced work travel [26]. Finally, one respondent highlighted the potential value of a recently published local framework that provides guidance for communities in embedding environmental and policy changes and subsequently enabling residents to take up positive behaviours [27].

Possible future research identified by panellists included exploring how the uptake of energy-efficient home improvements could be further promoted, particularly within the private rental sector. In this context, modelling different scenarios of home retrofit subsidy programmes was mentioned as a possible approach. Furthermore, one expert participant suggested employing large-scale surveys with homeowners from different socio-demographic categories, which could help identify relevant attitudes and values impacting their decision making on these issues. More generally, discussions highlighted that working with UK housing associations will be vital when designing interventions to address heat behaviours at scale.

#### 3.4.2. Consumption Behaviours (Eating Healthy/Eating More Fruit and Vegetables; Moving Away from Meat/Emission Heavy Diets)

During the discussion of consumption behaviours, panellists agreed that the behaviour ‘moving away from meat and dairy consumption’ was the most important out of all the identified target behaviours. Attendees suggested that this was due to the well-established link between meat and dairy consumption and high emission production and the health benefits of plant-based/higher vegetable-sourced diets (as the latter behaviour has a smaller carbon footprint in comparison to meat and dairy-focused diets). The identified barriers to the uptake of this behaviour related to issues with public acceptability of state involvement in dietary changes as well as issues with accessibility of alternatives to high-emission diets. On the other hand, the group felt that monetary incentivisation could facilitate the adoption of a low-emission diet, for example, by increasing the price of high-emission food items and lowering the cost of low-emission foods. Finally, the group suggested that subsidisation may be an effective measure for encouraging food producers to shift towards lower-emissions produce.

Potential future research for this behaviour is largely focused on promoting public acceptability of a reduction in meat consumption and exploring nutritional awareness, with a focus on school-aged children. There was also interest in exploring menu changes in easily controllable settings, particularly school and work canteens. Some expressed a perceived need to evaluate existing behavioural interventions such as food labelling systems, taxation and subsidisation. Furthermore, panellists highlighted that research should be conducted on quantifying the impact of potential emissions savings to ensure alignment with emissions reduction targets as outlined by the Committee on Climate Change.

#### 3.4.3. Transport Behaviours (Reducing Number of Cars Per Household; Walking Shorter Journeys; Reducing Day/Weekend Leisure Car Journeys)

The discussion about the target behaviours was largely contextualised as promoting active travel as an alternative to car-based travel. The behaviour, ‘reducing car use and ownership’ was particularly discussed in conjunction with its positive effect on the increasing use of public transport (as an alternative personal car usage and ownership), and how the resulting reduction of car usage could lead to reductions in emissions. This behaviour was considered of great importance given the perceived scarcity of interventions to promote the use of public transport. Some panellists highlighted that future research should recognise regional differences in public transport infrastructure capability between rural and urban areas. In relation to COVID-19, workshop attendees reflected that people may have been discouraged from using public transport during the pandemic due to the heightened risk of spreading the virus in crowded and poorly ventilated spaces. Panellists speculated that this trend has likely motivated the increased use of private cars for travel, thus further justifying continued car ownership. One respondent highlighted the importance of reflecting on the effect of active travel practices in other countries, for example, the Netherlands, where despite the common use of bicycles as means of transport, emissions are still similar to the UK, suggesting that an increased uptake in active travel behaviours has not necessarily reduced car usage [28]. This emphasises the importance of continuously monitoring the effects of increased active travel on emission levels.

The group identified that the main barriers to adopting active travel related to ‘land-locking’; that is, that housing is often situated long distances from essential amenities, access to public transport, and active travel infrastructure. On the other hand, key facilitators related to monetary incentivisation (in terms of subsidising public transport, which was expected to encourage moving away from personal car usage), and workplaces taking a role in promoting active transport (for example, by offering cycle to work schemes to help make active travel a more accessible and convenient option for commuters). Referenced literature included a case study examining the effects of increased active travel on climate change mitigation in the Netherlands [28], a regional transport strategy for Swindon, Oxfordshire, Cambridgeshire, Northamptonshire and Hertfordshire geographical regions to inform future sustainable transport infrastructure investment [29], and a single study examining barriers and facilitators to the uptake of active and public transport in a metropolitan setting [30].

### 3.5. Cross-Cutting Ideas

There were several research areas that cut across the target behavioural categories, particularly when viewed through the lens of COVID-19. Firstly, panellists expressed a desire to understand the extent to which the public would be open to continuing significant lifestyle changes implemented during COVID-19 for the benefit of the environment. For example, would individuals be willing to continue working from home long term in a post-COVID-19 world? Indeed, the importance of identifying methods to maximise emissions reduction as we transition into a ‘home-working’ society (taking into consideration mental health and productivity) was also discussed. More generally, panellists agreed that there is a need for further integrated research focusing on the co-benefits of climate and health. Furthermore, attendees suggested that future frameworks or tools in this area should ensure that climate interventions or policies are equitable and inclusive. Finally, the expert panel reflected that while policy change is more relevant for encouraging long-term/grand-scheme national level/wider population behaviour change, social norms should also be addressed.

**Table 5 ijerph-20-05094-t005:** Future research ideas relating to prioritised target behaviours proposed by the expert panel during the final discussion session of the workshop.

Target Behaviour	Proposed Research Questions	Additional Literature
Heat (installing double/triple glazing; installing cavity wall insulation; installing solid wall insulation)	- How do you ensure that private rented sector is not left behind from energy efficiency/ home improvement policies? - Which factors influence house owners’ decisions in relation to home improvements, and what motivates them to prioritise energy efficiency? - How could the uptake and attractiveness of heat pumps be increased? - How does a retrofit support scheme need to be designed to be acceptable (e.g., What level of accessibility, funding and resource is required)?	Lessons from energy efficiency policy and programmes in the UK from 1973 to 2013, Mallaburn and Eyre [25] The appeal of the green deal: Empirical evidence for the influence of energy efficiency policy on renovating homeowners, Pettifor, Wilson [24] Planning for sustainable growth in the Oxford-Cambridge Arc: an introduction to the spatial framework [27] A systematic review of the energy and climate impacts of teleworking, Hook, Sovacool [26] Centre for Research into Energy Demand Solutions: Publications repository
Consumption (Eating healthy/eating more fruit and vegetables; moving away from meat/emission heavy diets)	- Exploring the acceptability of reducing meat and dairy food options in controllable environments (schools, nurseries, work canteens) - Citizen’s juries exploring publicly acceptable options for promoting moving away from meat & dairy. - School pilot studies to explore child acceptability, nutrition, and awareness of plant-based diets - Comparison of the equity implications of food labelling of plant-based products and meat/high-emissions products.	Review: Demand-Side Food Policies for Public and Planetary Health [31], Dietary Patterns for Health and Sustainability: From Experts’ Opinions to Action for the WHO European Region, Bach-Faig, Wickramasinghe [32] Health impacts and environmental footprints of diets that meet the Eatwell Guide recommendations: analyses of multiple UK studies, Scheelbeek, Green [33]
Transport (Reducing number of cars per household; Walking shorter journeys; Reducing day/weekend leisure car journeys)	- Pilot workplace incentivisation programmes for active travel, e.g., free transport passes, bonuses, bike storage. - How can we change travel in rural areas where distances are long and public transport sparse? - What are the differences in climate/emissions reductions impacts of adopting active transport (versus personal car usage) between walking shorter journeys and walking longer journeys?	Individual carbon dioxide emissions and potential for reduction in the Netherlands and the United Kingdom, Susilo and Stead [28] Urban Form and Trends of Transport Emissions and Energy Consumption of Commuters in the Netherlands, Susilo and Stead [34] Regional Transport Strategy, England’s Economic Heartland [29] Achieving recommended daily physical activity levels through commuting by public transportation: unpacking individual and contextual influences, Wasfi, Ross [30]
Cross-cutting	- What communications/ behavioural change lessons from the pandemic can be utilised for to promote pro-environmental behaviour change? - What are the critical policy-behaviour ‘windows of opportunity’ that exist now (during COVID-19 and lockdown but may not later) for making change? E.g., shifts to travel regulations, land planning requirements, etc. - Identifying long-term impacts of the pandemic and what means for emissions and policies (e.g., employment, economy, health, education, engagement, disaster fatigue) - What are the trade-offs between increased home working vs. reduced transport? (and energy use in workplaces	CAST briefings https://cast.ac.uk/publications/briefings/, accessed on 21 April 2022. Net Zero Societal Change Analysis Project [18]

## 4. Discussion

A Delphi study involving a total of 21 experts was conducted to identify and prioritise climate change mitigation behaviours with the aim of identifying areas ripe for further research within the context of climate change mitigation, considering the extended impact of the COVID-19 pandemic on behaviour change. The Delphi study process prioritised the following climate change mitigation behaviours: Three behaviours from the category heat (installing double/triple glazing; installing cavity wall insulation; installing solid wall insulation), one behaviour from the consumption category (eating healthy/eating more fruit and vegetables; moving away from meat/emission heavy diets) and three transport behaviours (reducing number of cars per household; walking shorter journeys; reducing day/weekend leisure car journeys). The behaviours are broadly consistent with previous research that identified behaviours that contribute to anthropogenic climate change, therefore presenting priority areas for intervention [7].

Furthermore, experts particularly highlighted practical barriers that impacted people’s ability and motivation to engage in identified key behaviours, particularly financial barriers. These are typically related to the commonly higher costs required to perform the more sustainable behaviours, in combination with a lack of policy-regulated subsidies designed to ensure affordability for members of the public, as well as a lack of infrastructure. This holistic point of view of attributing behaviour not only to psychological factors but also to the wider environment aligns with the Behaviour Change Wheel, a commonly used, evidence-based behaviour change framework that proposes three categories to explain behaviours, including capability (C), opportunity (O) and motivation (M), also referred to as COM-B [35]. Capability can be distinguished between the physical and psychological capability to perform behaviours (“psychological capability being the capacity to engage in the necessary thought process—comprehension, reasoning.” [35]). Motivation involves all reflective and automatic processes (“involving emotions and impulses that arise from associative learning and/or innate dispositions” [35]). Finally, with regard to opportunity, we distinguish between “physical opportunity afforded by the environment and social opportunity afforded by the cultural milieu that dictates the way we think about things” [35]. This is consistent with a recently published meta-analysis that concluded effective behaviour change interventions in the context of climate change mitigation involve a combination of education/information and choice architecture interventions [7].

While the workshop discussion identified frameworks and additional literature that provide some insights into the existing interventions and guidance and broadly address the prioritised target behaviours, little evidence could be identified with regard to effective intervention features. Nonetheless, a number of potentially promising approaches were discussed in relation to selected prioritised behaviours, including: menu changes in easily controllable settings (e.g., schools, work canteens), the use of food labelling systems, taxation and subsidisation in the context of consumption behaviours. These potential approaches will all, however, require further evaluation to assess their long-term effects. Where heat behaviours were concerned, the Green Deal was discussed but was established as a failed initiative despite its potential appeal to private homeowners, because of the relatively slow uptake to date due to homeowners’ uncertainty about financial benefits [24]. More generally, panellists stressed the importance of adopting an interdisciplinary and inclusive approach during the development of interventions to ensure the learning and integration of relevant disciplines within climate change, behavioural science and public health. The importance of integrating stakeholders (e.g., policy decision-makers, target groups of the relevant behaviours) early on was also emphasised to ensure interventions are both attractive and feasible for the public, meaning that the climate-friendly action should be the easier option for the target group in order to mitigate any practical barriers such as limited accessibility or cost [7].

Finally, while experts identified several climate mitigation behaviours as being impacted by COVID-19 based on the submitted responses from the questionnaires, there was surprisingly little focus on COVID-19 during the workshop discussions in the context of the behaviours that experts decided to focus on. This provides limited conclusions about the direction and future impact of COVID-19 on climate change. The main reflection emerging from the workshop related to the need for continued monitoring of behaviour shifts as a result of COVID-19 to help identify potential long-term impacts on people’s lifestyles, for example, a continued trend towards working from home resulting in reduced car-based travel. Recent data demonstrate that the global response to the COVID-19 pandemic and subsequent behavioural shifts led to a sudden reduction in both GHG emissions in 2020; however, these positive effects are close to negligible, and lasting effects, if any, will only arise via a green recovery strategy. Although there is a high risk of rebounding to previous, unsustainable pathways, COVID-19 should be seized as a window of opportunity to trigger policies and shift norms that could contribute to decarbonisation over the long term if these can be sustained post-pandemic [17,36].

### 4.1. Recommendations for Further Research

The expert panel discussion identified a range of opportunities for future research, which can be categorised into two broad themes, including (a) explicit and discrete recommendations for each of the specific behaviours discussed in the workshop, and (b) broader recommendations. These are elaborated further in the following sections.

### 4.2. Further Research on Prioritised Target Behaviours

Research ideas for the heat behavioural category included exploring the barriers and facilitators of retrofitting installation and installing heat pumps to improve the acceptability of said schemes and understanding how the private rented sector can be included in energy efficiency policies. Research ideas for the consumption behavioural category included plant-based food pilot studies in controllable environments such as schools and workplace canteens, a comparison of the equity implications of food labelling of plant-based products and meat/high-emissions products and identifying publicly acceptable options for meat and dairy alternatives. Lastly, research ideas for the transport behavioural category involve an improved understanding of how travel can be provided in rural areas and areas with sparse public transport. Examples of future research may include examining the differences in emissions between adopting public transport versus walking journeys and piloting workplace-based interventions to promote active travel. Cross-cutting behavioural research topics included identifying the long-term impacts of COVID-19 on behaviour and climate change mitigation. Specific research questions the panel identified in relation to each target behavioural category are listed in Table 5.

Furthermore, the relationship between some behaviours and their health impacts was underexplored in discussions. Therefore, further research should focus on exploring the health and climate change co-benefits of mitigation behaviours and interventions [7,37].

### 4.3. Further Research on Climate Change Mitigation

Firstly, the findings from the Delphi study suggest that all identified climate mitigation behaviours need to be investigated in more detail in the future, with a particular focus on identifying effective interventions that are designed to overcome various sources of barriers, including knowledge, motivational, and practical barriers. More generally, the Delphi study further supports the importance of emphasising the co-benefits of climate change, which could present a promising opportunity to engage different types of people from the public by appealing to different values and beliefs for motivating climate change action (such as health), which could be addressed in future interventions. Appealing to a wider range of audiences through tailored interventions may therefore ultimately enable the adoption of climate change mitigation behaviours on a wider scale.

Furthermore, as mentioned previously, the Delphi study reinforces the importance of addressing practical barriers in order to ensure accessibility to behaviours in the public as a necessary basis, considering that financial and environmental barriers (e.g., lack of affordability policies and infrastructure) were perceived as limiting factors to people’s ability to adopt and sustain recommended behaviour changes, despite existing awareness and a general willingness to adopt changes in the public. Indeed, a need for economic incentives to remove behavioural barriers to climate behaviour adoption aligns with findings from two literature reviews [38,39]. For most behaviours, including all prioritised behaviours in the Delphi study, this will require a multidisciplinary approach that will require close collaboration with policy stakeholders whose buy-in and engagement are vital to advocate for and embed relevant policies, subsidies, pricing and schemes to ensure that the behaviour change measures reach the public and result in the desired impact.

Finally, a third recommendation resulting from the present work relates to an increased need for more rigorous evaluations of behaviour change interventions that address climate change mitigation. This will help to strengthen the evidence base in light of the currently limited understanding of effective approaches that result in quantifiable, impactful changes [7]. Given the key points discussed in the workshop, it is particularly important to consider public accessibility and acceptability, in addition to the overall contribution of specific interventions in reducing greenhouse gases, which highlights the scope for future research. Key gaps in the literature include a lack of studies conducted using randomized controlled trials or follow-ups and quantifying the effect of the three identified behavioural areas (i.e., reducing driving, meat consumption and household energy use) on emissions reductions [40].

### 4.4. Limitations

Despite the positive outcomes from the workshop in terms of identifying areas ripe for further research within the context of climate change mitigation, there were some limitations that should be considered when interpreting the results from this study. Firstly, there were difficulties identifying and recruiting climate change experts who specialise in public health. The recommended number of expert panellists for Delphi studies ranges between 10 and 50, depending on the topic area and researcher resources [23]. While the number of expert panellists in this study falls within this range, there was a small imbalance of experts who informed the Delphi study and workshop discussion, resulting in a slightly higher representation of behavioural science experts compared to public health experts. This imbalance may have contributed to the reduced discussion of health aspects as reflected in the preceding section. Secondly, the attrition of experts in Round 3 of the Delphi study was high (it was difficult to arrange a suitable date for the workshop for all experts that participated in the preliminary rounds of the study), and so only 9 out of the total 21 participants contributed to the third questionnaire iteration. Due to issues with attrition between subsequent Delphi rounds, we were unable to measure the consistency of responses (as recommended by Nasa, Jain and Juneja [41]) to utilise as an additional stopping criterion. Finally, although the purpose of a Delphi exercise is not to examine every topic exhaustively but to be led by the expert panel to focus on relevant areas for discussion, due to time constraints during the workshop we were unable to address some research questions in detail, particularly regarding the nature and extent of the impact of COVID-19 on relevant behaviours. Furthermore, although the study measured whether panellists viewed behaviours as being impacted positively or negatively by COVID-19 (Round 1 behaviours and Round 2 additional behaviours), this distinction was not reflected specifically in subsequent rounds of the Delphi study and was amalgamated to identify whether behaviours had been impacted by COVID-19. This was carried out for simplicity, as the additional rating of positive/negative was not relevant to the interpretation of the scores. Research in the near future should: explore whether climate change mitigation behaviours impacted by COVID-19 have overall resulted in positive or negative consequences for the climate (which would take more time and a dedicated study with appropriate methods to monitor); therefore, in hindsight, this aspect was beyond the scope of this study. Furthermore, research in the near future should aim to explore to what extent current behavioural shifts as a result of COVID-19 can be sustained post-pandemic.

## 5. Conclusions

The present study aimed to identify and prioritise key target behaviours according to climate change and health impacts, to understand how these behaviours might be affected by COVID-19, and to examine the current evidence base to inform the design of future interventions. A total of seven behaviours were identified as target behaviours: three heat-related behaviours (installing double/triple glazing; installing cavity wall insulation; installing solid wall insulation), one consumption-related behaviour (eating healthy/eating more fruit and vegetables—moving away from meat/emission heavy diets); and two transport-related behaviours (reducing the number of cars per household; walking shorter journeys; reducing day/weekend leisure car journeys). Future research should increasingly explore how health co-benefits could be exploited to maximise the adoption of the identified behaviours and focus on the design and evaluation of interventions to address the identified key behaviours, considering relevant practical, knowledge and motivational barriers.

## Figures and Tables

**Figure 1 ijerph-20-05094-f001:**
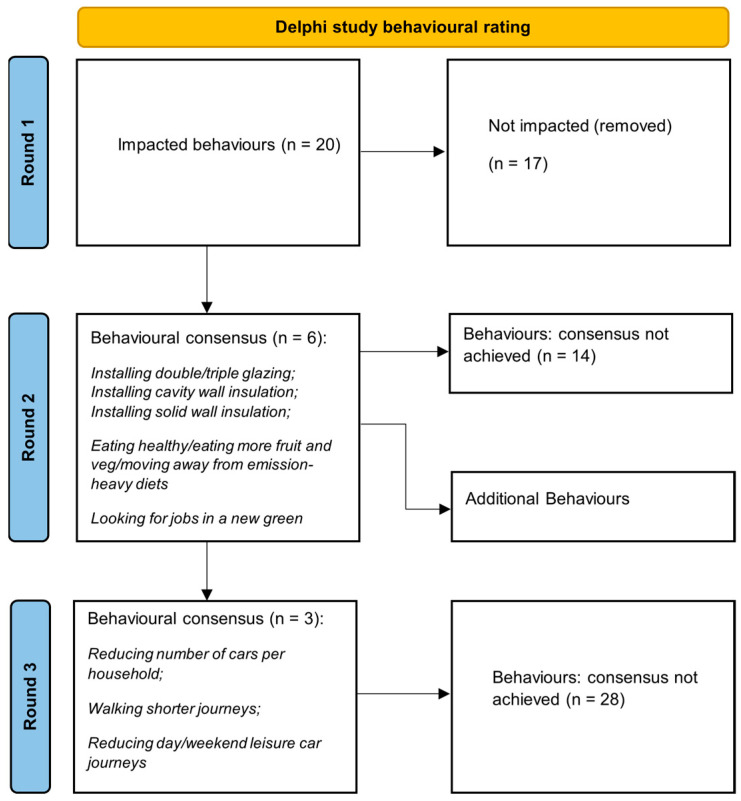
Delphi study Round 1–3 behaviour inclusion and exclusion.

**Table 1 ijerph-20-05094-t001:** Climate change mitigation behaviours identified as ‘impacted’ by COVID-19 in Round 1 (over 50% of experts indicated behaviour has been impacted), behaviours removed/not progressed to Round 2 (below 50% of experts indicated behaviour has been impacted by COVID-19) and percentage value of panellists that indicated respective behaviours were impacted by COVID-19.

Behaviour	Impact (Percentage %)
Heat	
Impacted Behaviours	
Reducing number of rooms heated	95.2
Heating for fewer hours of the day	95.2
Reducing thermostat temperature	81
Cooking shorter meals or meals in bulk	66.7
Installing double/triple glazing	61.9
Installing cavity wall insulation	57.1
Installing solid wall insulation	57.1
Installing thermostatic radiator valves (TRVs)	57.1
Removed Behaviours	
Cooling house by opening windows	47.6
Rinsing the dishes and washing hands in cold water	47.6
Taking up a service-based heat proposition	47.6
Connecting to district heat network (DHN)	33.3
Transport	
Impacted Behaviours	
Reducing number of air miles	100
Taking fewer holidays	100
Combining trips	76.2
Walking or cycling to school	71.4
Reducing number of cars per household	52.4
Removed Behaviours	
Living closer to work and amenities	47.6
Extending how long a car is used for	38.1
Buying and using a smaller car	35
Plugging in electric vehicles whenever possible and accepting smart charging	23.8
Consumption	
Impacted Behaviours	
Reducing waste food	95.2
Eating healthy/eating more fruit and vegetables (moving away from meat/emission heavy diets)	61.9
Removed Behaviours	
Buying sustainable products	33.3
Consumers purchasing informed by strength of companies/manufacturers carbon footprint	20
Electricity	
Removed Behaviours	
Installing LED lighting	28.6
Buying smart-ready appliances	23.8
Switching to time-of-use tariff	33.3
Charging electric vehicle at home (smart charging, V2G, etc.)	28.6
Waiting for a full load before using washing machine	47.6
Doing dishes by hand	28.6
Buying a smaller refrigerator	20
Non-Sectoral/Societal Change	
Impacted Behaviours	
Improving home workspace	100
Transitioning to digital working	100
Looking for jobs closer to home	90
Looking for jobs in a new green economy	52.4
Informed property purchasing	47.6

**Table 2 ijerph-20-05094-t002:** Climate change mitigation behaviours that achieved and did not achieve consensus in Round 2, climate change impact Likert importance scores, health impact criteria Likert importance scores, cumulative importance scores (TOTAL score), climate change impact IQR values (CCI IQR) and health impact IQR values (HI IQR).

Behaviour	Climate Change Impact	Health Impact	TOTAL Score	CCI IQR	HI IQR
Heat					
Installing double/triple glazing *	5	3	8	1	1
Installing cavity wall insulation *	5	3	8	1	1
Installing solid wall insulation *	5	3	8	1	1
Installing thermostatic radiator valves (TRVs)	4	2	6	1	2
Cooking shorter meals or meals in bulk	3	1	4	1.5	0.5
Reducing thermostat temperature	4	3	7	2	3
Heating for fewer hours of the day	4	2	6	2	2.75
Reducing number of rooms heated	4	2	6	2	2.5
Consumption					
Eating healthy/eating more fruit and vegetables (moving away from meat/emission heavy diets) *	5	5	10	0.5	0
Reducing food waste	4	2	6	1	0
Transport					
Reducing number of cars per household	4	4	8	1	3
Walking or cycling to school	4	5	9	1.5	1
Reducing number of air miles	5	2	7	0	3
Taking fewer holidays	3	1	4	3	2
Combining Trips	4	1	5	2	1.5
**Non-sectoral/Societal Change**					
Looking for jobs in a new green economy *	4	2	7	1	1
Looking for jobs closer to home a	4	3	7	1	1
Improving home workspace	2	4	6	1	2
Informed property purchasing	3	2	5	2	1.5
Transitioning to digital working	3	2	5	2	2

* The superscript denotes behaviours that reached consensus (IQR score ≤ 1 for respective climate change impact criteria and health impact criteria.

**Table 3 ijerph-20-05094-t003:** Additional climate change mitigation behaviours (identified in Round 1 and rated in Round 2) identified as ‘impacted’ by COVID-19 (over 50% of experts indicated behaviour has been impacted), behaviours removed/not progressed to Round 3 (below 50% of experts indicated behaviour has been impacted by COVID-19) and percentage value of panellists that indicated respective behaviours were impacted by COVID-19.

Behaviour	Percentage Impact %
Heat	
Impacted Behaviours	
Reducing usage of open fire/wood burning stoves	63.2
Reducing hot water usage for washing/showering (hygiene behaviour)	63.2
Taking less showers	57.9
Removed Behaviours	
Installing a heat pump	36.3
Wearing warmer clothes	42.1
Transport	
Impacted Behaviours	
Walking or cycling to shops	94.7
Increasing use of public transportation	94.7
Reducing single-occupancy car use	94.7
Reducing car commuter journeys	94.7
Reducing day/weekend leisure car journeys	94.7
Purchasing a bicycle	84.2
Increasing domestic holidays (taking less international travel holidays)	89.5
Walking shorter journeys	89.5
Consumption	
Impacted Behaviours	
Reducing internet usage	100
Reducing use of online shopping/e-commerce services	89.4
Purchasing fewer new electronic goods	79
Purchasing fewer new clothing	79
Purchasing home improvement items	72.2
Increasing healthy and sustainable eating by growing own produce	68.4

**Table 4 ijerph-20-05094-t004:** Climate change mitigation behaviours that achieved consensus in Round 3 (including additional behaviours), climate change impact Likert importance scores, health impact criteria Likert importance scores, cumulative importance scores (TOTAL score), climate change impact IQR values (CCI IQR) and health impact IQR values (HI IQR).

Behaviour	Climate Change Impact	Health Impact	TOTAL Score	CCI IQR	HI IQR
Transport					
Reducing number of cars per household *	2	4	6		
Transport Additional Behaviours					
Walking shorter journeys *	4	4	8	0.25	0.5
Reducing day/weekend leisure car journeys *	4	2	6	1	1

* The superscript denotes behaviours that reached consensus (IQR score ≤ 1 for respective climate change impact criteria and health impact criteria.

## Data Availability

The data presented in this study are available in within the article.

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
