# Peer review of "Prioritising Climate Change Mitigation Behaviours and Exploring Public Health Co-Benefits: A Delphi Study"

_ijerph, 2023, doi:10.3390/ijerph20065094_

Round 1
Reviewer 1 Report
This paper presents climate change mitigation behaviours using Delphi methodology.
In this study total 21 participants contributed, the number of contributions may be increased.
How IQR score calculated may be explained.
Author Response
Thank you for your comments, we have addressed your comments as follows:
This paper presents climate change mitigation behaviours using Delphi methodology. In this study total 21 participants contributed, the number of contributions may be increased.
The recommended number of expert panellists recommended to participate in Delphi study methodology ranges between 10 – 50. Our study involving 21 participants sits within this range however, we acknowledge in the limitations that since we had two categories of experts (behavioural science & climate change, public health & climate change) and that there was an imbalance between these categories. We have discussed this further and included a reference to the expert panellist range and Delphi methodology in the updated manuscript.
How IQR score calculated may be explained.
I have provided an explanation of how IQR score was calculated.
Reviewer 2 Report
The study aimed to identify and prioritize key target behaviors according to climate change and health impacts in order to understand how COVID-19 might affect these behaviors.
First, we would like to note that 21 experts were involved in the study. We believe that this is an insufficient number of specialists for a correct analysis.
It is not clear how the reader can evaluate the qualifications of specialists. Perhaps the authors should add a table with a list of organizations and positions of specialists participating in the study.
In addition, it is not clear how the measures proposed by experts will reduce the carbon footprint (CO2e). We recommend adding numerical indicators tied to a specific year.
It is also not clear how the proposed measures will affect people's health (for example, giving up meat).
Now it is difficult to assess the significance of the work due to the lack of specific numerical indicators.
Author Response
Thank you for your comments. Please see our responses below:
The study aimed to identify and prioritize key target behaviors according to climate change and health impacts in order to understand how COVID-19 might affect these behaviors. First, we would like to note that 21 experts were involved in the study. We believe that this is an insufficient number of specialists for a correct analysis. It is not clear how the reader can evaluate the qualifications of specialists. Perhaps the authors should add a table with a list of organizations and positions of specialists participating in the study.
The recommended number of expert panellists recommended to participate in Delphi study methodology ranges between 10 – 50. Our study involving 21 participants sits within this range however, we acknowledge in the limitations that since we had two categories of experts (behavioural science & climate change, public health & climate change) and that there was an imbalance between these categories. We have discussed this further and included a reference to the expert panellist range and Delphi methodology in the updated manuscript. Due to anonymity agreements for study participation, we cannot include a list of organisations for the experts however, we have now included a list of types of panellists (senior academics, senior government researchers, etc.) and numbers of each for the respective expert groups.
In addition, it is not clear how the measures proposed by experts will reduce the carbon footprint (CO2e). We recommend adding numerical indicators tied to a specific year. It is also not clear how the proposed measures will affect people's health (for example, giving up meat). Now it is difficult to assess the significance of the work due to the lack of specific numerical indicators.
In terms of quantifying mitigation/carbon emissions from the adoption of the discussed behaviours, this is a complex task due to the broadness of behaviour types and contexts. Quantifying impacts is out of scope for this paper, referenced literature throughout the manuscript establish the effectiveness of discussed behaviours to mitigate climate change. The primary focus of paper is overarching health protection via climate change mitigation versus health improvement (focus is health co-benefits broadly versus focusing specifically on what these health benefits are- we already know from the literature that there are established health benefits). Furthermore, we have addressed in the limitations that due to an imbalance of experts (greater number of behavioural science experts than public health experts) that there was unfortunately a reduced discussion of health impacts at the stage of the Workshop due to attrition.
Reviewer 3 Report
Thank you for the opportunity to review the manuscript “Prioritising climate change mitigation behaviours and exploring public health co-benefits: A Delphi study”.
The paper describes the process conducted to understand and prioritise specific climate change mitigation actions that also positively impact health outcomes, and which of these has been influenced by behaviour changes associated with COVID-19 in the UK. This is a worthwhile and useful piece of work which can help influence policy and research directions in climate change mitigation behaviours.
The following points are recommended for review by the authors.
Introduction
Lines 78-80: It would be useful here to also include reference to the body of research that shows how discussing climate change through a health lens can help shift individuals across the spectrum from denial to action, eg. Maibach et al. 2010 https://bmcpublichealth.biomedcentral.com/articles/10.1186/1471-2458-10-299
Lines 90-95: It would be helpful to specify that this work is UK-based only.
Methods
Line 101: I am not clear of the meaning of the sentence “Consensus is reached by expert ranking processes several iterative rounds.” Missing a word?
Results
The general presentation and description of results needs significant overhaul. Clarity is needed around the following points.
Line 242: The term ‘percentage impact values’ is confusing. The phrase tends to imply that each behaviour was rated in terms of *how* positive or negative the impact was, rather than on a 3-point scale (positive, negative or no impact). The percentage rating appears to be derived from the percentage of panel members that rated the behaviour as having an impact (either positive or negative) (lines 208-209)? Please use an alternate label here, as this feeds through to the understanding of later rounds. Update this label in the Supplementary data.
The treatment of the 20 additional behaviours (line 249) is quite confusing. They appear to be identified in Round 1, but it is not clear where they are rated for their ‘percentage impact’. If in Round 1, should they not appear in Table 1/Round 2 section? If in Round 2, should they not appear in Table 1/Round 3 section? They are currently hidden in Supp data but appear important to the narrative and need to be moved to the main paper.
How are the Table 1 columns ‘climate change impact’ and ‘health impact’ calculated? In the text these are described as ‘individual importance scores’ (line 265). The methods section only discusses the calculation and importance of the IQR (Section 2.3.2).
Table 1 results are confusing as an amalgamation of Rounds 1-3. Separate tables on the outcomes of each round (e.g. matching Fig 1 outcomes by Round) would make this much clearer.
Grouping positive and negatively impacted behaviours together is a little confusing. I would prefer to see an additional column to show which were positive and which were negative, or alternatively, grouped by positive/negative.
Discussion
The discussion does not adequately cover the importance of these findings in the context of COVID-19, which appears to be the main aim of the research. Instead, the discussion centres on the behaviours as they related to mitigation and health impacts. An additional paragraph on this topic would be useful.
Supplementary data
Update labels as above
Amend heading for section 3 to more accurately reflect how the percentages are derived
Spelling mistake in section 3 heading (ptogressed)
Author Response
Thank you for your comments. Please see our responses below:
Introduction
These requested edits have been implemented in the specified sections.
Results
To address the concerns raised by the use of the term ‘percentage impact values’, we have re-worded in the text to clarify that values refer to the percentage of panellists that indicated the respective behaviour was impacted by COVID-19. We have moved tables presenting impact % values from the supplementary materials into the manuscript but have not presented separate percentage values for negative and positive values for each behaviour- we have addressed our reasons for this in the Methodology and Limitations.
Discussion
A paragraph has been added to reflect the implications related to COVID-19, which the limitations of making conclusions due to the limited focus on this topic during the workshop (determined by participants’ decision on topic areas), also providing implications for future research to address the limited insights from the current study.
Supplementary data
Labels/table titles have been updated and moved to the manuscript and spelling mistake has been corrected.
Round 2
Reviewer 2 Report
Accept in present form.
Author Response
Thank you for your response. Could you please specify in what way you think the cited references in our manuscript should be improved? I have embedded all changes from the first round of reviewer comments into the manuscript if this easier to navigate, thank you.
Reviewer 3 Report
Thank you for your edits and additions. The additional information provided on the impact of COVID-19 seems especially relevant to the paper. An additional sentence in the Abstract to cover this finding would be appropriate, and match the stated aim of the research.
Author Response
Thank you for your response, we have now amended the Abstract to cover some conclusions about COVID (have maintained word count of 200)- please see line in red text.